



# Fluorescence properties of long-range transported smoke: Insights from five-channel lidar observations over Moscow during the 2023 wildfire season

Igor Veselovskii[1], Mikhail Korenskiy[1], Nikita Kasianik[1], Boris Barchunov[1], Qiaoyun Hu[2], Philippe Goloub[2], Thierry Podvin[2]

[1]Prokhorov General Physics Institute, Vavilova str.38, Moscow, Russia.
[2]Univ. Lille, CNRS, UMR 8518 - LOA - Laboratoire d'Optique Atmosphérique, F-59650 Lille, France

*Correspondence to*: Qiaoyun Hu (qiaoyun.hu@univ-lille.fr)

**Abstract.** The fluorescence lidar at the Prokhorov General Physics Institute (Moscow) was utilized to study transported over the Atlantic smoke during the wildfire season from May to September 2023. The lidar system, which is based on a tripled Nd:YAG laser, performs fluorescence measurements across five spectral intervals centered at wavelengths of 438, 472, 513, 560, and 614 nm. This configuration enables the assessment of the spectral dependence of fluorescence backscattering over a broad range of altitudes, from the planetary boundary layer (PBL) to the upper troposphere and lower stratosphere (UTLS). The fluorescence capacity of smoke, defined as the ratio of fluorescence backscattering to aerosol backscattering at the laser wavelength, exhibits significant variation in the UTLS, with changes of up to a factor of 3. This variation is likely indicative of differences in the relative concentration of organic compounds within the smoke. Analysis of more than 40 smoke episodes has enabled an evaluation of the height dependence of smoke fluorescence properties. Observations reveal that the fluorescence capacity generally increases with altitude, suggesting a higher concentration of organic compounds in the UTLS compared to the lower troposphere. Additionally, the measurements consistently show differences in the fluorescence spectra of smoke and urban aerosol. Urban aerosol fluorescence tends to decrease gradually with wavelength, whereas the peak of smoke fluorescence is observed at the 513 nm or 560 nm channels. This spectral distinction provides an effective means of separating smoke from urban aerosol. The technique was applied to analyze events where smoke from the upper troposphere descended into the PBL and mixed with urban particles, demonstrating its utility in distinguishing between these aerosol types.

## 1 Introduction

Smoke particles produced by intense wildfires can ascend into the upper troposphere and lower stratosphere (UTLS), where they can alter the Earth's radiation budget by scattering and absorbing solar radiation, as well as by influencing cloud formation processes (Baars et al., 2019; Khaykin et al., 2020; Senf et al., 2023). Mie-Raman lidars have been used for years to study the physical properties of both fresh and aged smoke (Adam et al., 2020; Ansmann et al., 2021, and references therein). However, distinguishing smoke from other aerosol types based on Mie-Raman lidar measurements can be challenging, especially within



thin aerosol layers where lidar ratios cannot be calculated. A new opportunity for identifying smoke particles has recently
emerged due to advancements in fluorescence lidar observations (Reichardt et al., 2018, 2023; Veselovskii et al., 2020, 2024;
Hu et al., 2022; Miri et al., 2024). Organic compounds, which account for a substantial fraction of wildfire emissions, exhibit
high fluorescence cross-sections, providing a basis for monitoring smoke in the UTLS and within cirrus clouds (Veselovskii
et al., 2022a; Gast et al., 2024).

Measuring the total fluorescence spectrum (Sugimoto et al., 2012; Saito et al., 2018; Reichardt et al., 2023) offers a clear
advantage for aerosol identification. Nevertheless, single-channel fluorescence lidars are now developing due to their
simplicity. Furthermore, selecting the spectral interval for fluorescence observations with an interference filter allows for high
detection sensitivity, as modern interference filters have a transmission exceeding 90%. In a standard Mie-Raman lidar, which
employs a tripled Nd:YAG laser and measures the so-called $3\beta+2\alpha$ data set (comprising three backscattering and two aerosol

extinction coefficients), a segment of the spectral range, roughly between 415 and 525 nm, can be utilized for fluorescence
measurements. Recent studies have demonstrated that combining single-channel fluorescence and polarization measurements
enables the identification of major aerosol types, such as smoke, dust, urban particles, and pollen (Veselovskii et al., 2022b).
Additionally, the contributions of different particle types to the total backscattering coefficient of an aerosol mixture can be
evaluated (Veselovskii et al., 2024). However, particle classification using a single-channel fluorescence lidar is based on

fluorescence capacity (the ratio of fluorescence backscattering to aerosol backscattering at the laser wavelength), which
depends on relative humidity (RH). Particle hygroscopic growth diminishes fluorescence capacity, thereby affecting the
algorithm's accuracy.

Meanwhile, as recently shown by Veselovskii et al. (2023), hygroscopic growth does not affect the fluorescence spectrum.
Thus, fluorescence measurements at several discrete channels should mitigate the impact of relative humidity. Moreover, the

spectral dependence of fluorescence backscattering for specific aerosols can be used for their identification.

In 2022, a five-channel fluorescence lidar was put into operation at the Prokhorov General Physics Institute (PGPI) in Moscow
(Veselovskii et al., 2023). The year 2023 was characterized by intense wildfires in North America and Siberia, allowing to
study the fluorescence spectra of smoke across a wide range of altitudes, from the planetary boundary layer (PBL) to the UTLS.
This paper summarizes the results of fluorescence measurements of smoke layers over Moscow from May to September 2023.

We begin with a description of the lidar system in Section 2. In the first part of the results section (Section 3.1), we analyze
two typical observations in the UTLS and in the middle troposphere to identify the main features of the fluorescence spectrum.
Section 3.2 presents an analysis of over 40 smoke episodes to determine the mean spectral characteristics of fluorescence at
different altitude intervals. Section 3 introduces a novel approach to separating smoke and urban aerosol based on their
fluorescence spectra. This approach is applied to episodes where smoke layers transported over the Atlantic descended to the

PBL and mixed with urban particles. In the conclusion, we summarize our main findings.



## 2 Experimental setup and data analysis

Description of the 5-channels fluorescence lidar at PGPI, is given by Veselovskii et al. (2023). The lidar is based on a tripled Nd:YAG laser with pulse energy of 80 mJ at 355 nm and repetition rate of 20 Hz. Backscattered light is collected by a 40 cm aperture telescope and the lidar signals are digitized with Licel transient recorders with 7.5 m range resolution, allowing
simultaneous detection in the analog and photon counting modes. Measurements were performed through the laboratory window at an angle of 48 degrees to the horizon, which was the maximum available angle for observations. Additional atmospheric parameters were obtained from radiosonde measurements at the Dolgoprudny meteorological station, located about 50 km from the observation site.

The lidar operated at PGPI allows the detection of elastic and nitrogen Raman backscatter as well as fluorescence backscatter
within five spectral intervals. The central wavelengths and the widths of transmission bands of the fluorescence channels ($D_\lambda$) are as follows: 438/29 nm, 472/32 nm, 513/29 nm, 560/40 nm and 614/54 nm. The strong sunlight background during the daytime restricts the fluorescence observations to nighttime only. The aerosol extinction coefficients at 355 nm ($\alpha_{355}$) were computed from Raman observations as described by Ansmann et al., (1992). To calculate the backscattering coefficient ($\beta_{355}$) in the presence of clouds, this method was adapted as described by Veselovskii et al. (2022b).

To characterize aerosol fluorescence properties we use two key parameters: the fluorescence backscattering coefficient ($\beta_{F\lambda}$) and the fluorescence capacity ($G_{F\lambda}$) for each fluorescence channel centered at the wavelength $\lambda$. The $\beta_{F\lambda}$ is calculated from the ratio of the fluorescence backscattering and the nitrogen Raman backscattering, as described by Veselovskii et al. (2020). The $G_{F\lambda}$ is defined as the ratio of $\beta_{F\lambda}/\beta_{355}$. These parameters were utilized in previous studies with a single-channel fluorescence lidar (Veselovskii et al., 2020; Hu et al., 2022) and correspond to measurements integrated over the transmission band of each
channel. However, for the analysis of multichannel observations, where the width of the transmission band ($D_\lambda$) varies from one channel to another, both $G_{F\lambda}$ and $\beta_{F\lambda}$ should be divided by $D_\lambda$, for comparisons across channels. Thus, in this study we use the "spectral fluorescence backscattering coefficients", $B_\lambda = \dfrac{\beta_{F\lambda}}{D_\lambda}$, and the "spectral fluorescence capacity", $G_\lambda = \dfrac{B_\lambda}{\beta_{355}}$, resulting in units of $\text{Tm}^{-1}\text{sr}^{-1}\text{nm}^{-1}$ and $\text{nm}^{-1}$ respectively. For simplicity, these normalized properties are referred to as the "fluorescence backscattering coefficient", and "fluorescence capacity" throughout the paper. All profiles of particle properties
presented in this work were smoothed with the Savitzky – Golay method, employing a second order polynomial with 8 points in the window.

The aerosol volume ($V$) and mass ($M$) concentrations are crucial properties for characterizing smoke events. One of the simplest ways to derive particle volume concentration from lidar measurements is by employing the extinction-to-volume conversion factor $C_V^i$ for a specific type of aerosol, as suggested by Mamouri and Ansmann (2017), Ansmann et al. (2019,
2021), He et al. (2023). These conversion factors for smoke, dust and urban aerosol, derived from the AERONET observations at a wavelength of 532 nm, enable the determination of aerosol volume concentration from extinction measurements. For a specific aerosol type, the particle volume can be estimated using the formula:



$$V_i = \alpha_{532} \times C_V^i . \tag{1}$$

The indices $i=S, U, D$ correspond to smoke, urban, and dust respectively. In the UTLS, the profiles of extinction coefficients

cannot be calculated within the weak smoke layers, but the smoke volume concentration, $V_S$, still can be estimated from the

aerosol backscattering coefficient, assuming the lidar ratio ($S_{532}^S$) for smoke.

$$V_S = \beta_{532} \times S_{532}^S \times C_V^S \tag{2}$$

Based on the results presented in the publications by Mamouri and Ansmann (2017) and Ansmann et al. (2021) we use

conversion factor $C_V^S$ =0.13 µm³cm⁻³/Mm⁻¹ for smoke in our calculation. The lidar ratios of North American aged smoke at

532 nm have been reported in numerous publications (e.g. Burton et al., 2013; Haaring et al., 2018; Hu et al., 2022). In this

study, we use the mean value of $S_{532}^S$ =64 sr. Since our lidar system measures at 355 nm, the backscattering coefficient $\beta_{355}$

can be converted to the equivalent at 532 nm using the color ratio $CR=\beta_{355}/\beta_{532}$. Based on observations in Lille (Hu et al.,

2022), the $CR$ for aged smoke is about 2.2. The volume concentration can be recalculated to the mass using the smoke density

$\rho_S$=1.15 g/cm³ (Ansmann et al., 2021). Thus, the smoke volume concentration (in µm³/cm³) and mass concentration (in µg/m³)

can be estimated by multiplying $\beta_{355}$ (in Mm⁻¹sr⁻¹) by the factors 3.8 µm³cm⁻³/Mm⁻¹sr⁻¹ and 4.4 µgm⁻³/Mm⁻¹sr⁻¹ respectively.

The uncertainty of such estimations should be below 25%.

The partitioning of a three-component aerosol mixture in terms of backscattering coefficient was discussed in a recent

publication by Veselovskii et al. (2024). For a smoke-urban mixture, such partitioning is feasible when the fluorescence

capacities of smoke and urban particles, $G_\lambda^S$ and $G_\lambda^U$, are known. The equations for the backscattering coefficients and

fluorescence capacities can be expressed as follows:

$$\eta_S + \eta_U = 1 \tag{3}$$

$$G_\lambda = \eta_S G_\lambda^S + \eta_U G_\lambda^U \tag{4}$$

Here $\eta_{S,U} = \dfrac{\beta_{355}^{S,U}}{\beta_{355}}$ are the relative contributions of smoke and urban particles to the overall backscattering coefficient. From

Eq.3 and 4, we derive:

$$\beta_{355}^S = \frac{G_\lambda - G_\lambda^U}{G_\lambda^S - G_\lambda^U} \beta_{355} \tag{5}$$

Thus, the volume and mass concentrations of smoke in the smoke-urban mixture can be determined.

**3 Analysis of fluorescence measurements**



### 3.1 Fluorescence properties of smoke within the FT and UTLS

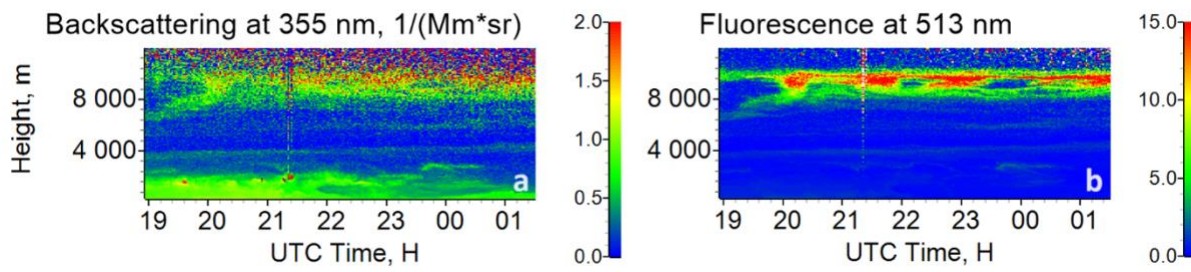

**Figure 1: Spatiotemporal distributions of (a) the aerosol backscattering coefficient $\beta_{355}$, and (b) the fluorescence backscattering coefficient $B_{513}$ (in $Tm^{-1}sr^{-1}nm^{-1}$) during the night of September 6-7, 2023.**

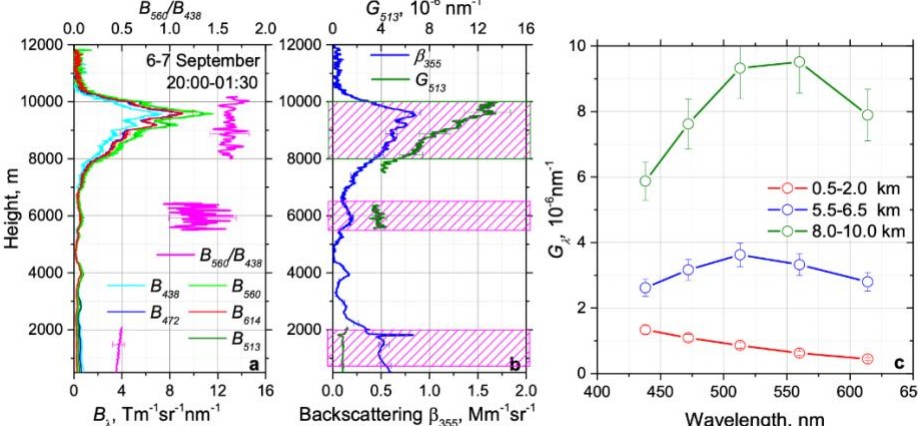

**Figure 2: Vertical profiles of (a) the fluorescence backscattering coefficients $B_\lambda$ at 438, 472, 513, 560, 614 nm along with the ratio $B_{560}/B_{438}$, and (b) the aerosol backscattering coefficient $\beta_{355}$ alongside the fluorescence capacity $G_{513}$. (c) Spectra of the fluorescence capacity $G_\lambda$ within three height ranges, which are marked on plot (b) by magenta boxes. Results are presented for the temporal interval from 20:00 to 01:30 UTC on the night of September 6-7, 2023.**

From May to September 2023, smoke layers originating from North American and Siberian wildfires were regularly observed over Moscow at a wide range of altitudes, extending from the PBL to the UTLS. In total, over 40 measurement sessions were analyzed. The fluorescence properties of smoke varied with height, and in this analysis, we considered three following altitude ranges: the PBL, the free troposphere (FT, 4-8 km) and the UTLS (8-12 km). The aerosols over the observation site were primarily composed of smoke and urban aerosols. In our classification scheme, the "urban" type encompasses both continental aerosol and anthropogenic pollution.

Fig.1 provides an example of a strong smoke episode in the UTLS during the night of September 6-7, 2023. The figure depicts spatiotemporal distributions of the aerosol and fluorescence backscattering coefficients ($\beta_{355}$ and $B_{513}$). The smoke layer within the 8-10 km range exhibits strong fluorescence with $B_{513}$ exceeding 10 $Tm^{-1}sr^{-1}nm^{-1}$ at its maximum. The mean value of the ratio $\dfrac{B_{513} \times D_{513}}{\beta_{355}}$ within this height range is about $2.7 \times 10^{-4}$, indicating that the fluorescence backscattering coefficient is almost




four orders of magnitude lower than the elastic one. However, the fluorescence technique in Fig.1 allows to monitor the smoke
layer in the UTLS.

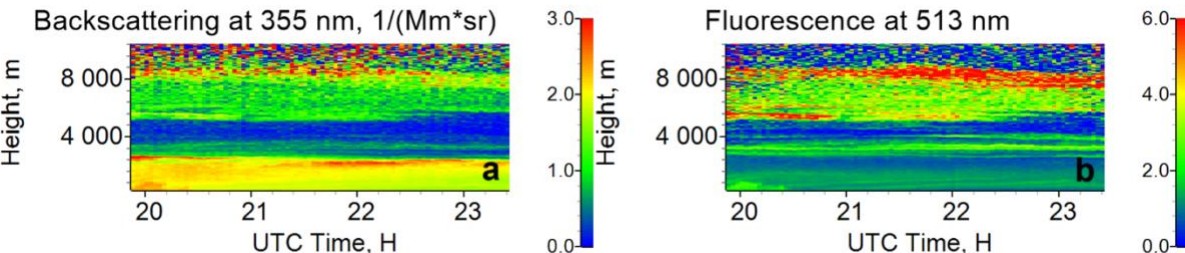

**Figure 3: Spatiotemporal distributions of (a) the aerosol backscattering coefficient $\beta_{355}$, and (b) the fluorescence backscattering coefficient $B_{513}$ (in Tm$^{-1}$sr$^{-1}$nm$^{-1}$) during the night of June 16, 2023.**

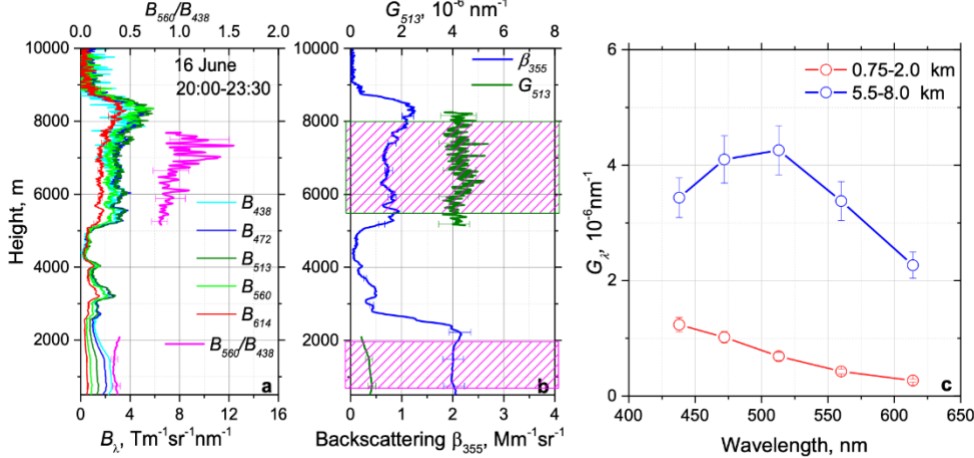

**Figure 4: Vertical profiles of (a) the fluorescence backscattering coefficients $B_\lambda$ at 438, 472, 513, 560, 614 nm along with the ratio $B_{560}/B_{438}$, and (b) the aerosol backscattering coefficient $\beta_{355}$ alongside the fluorescence capacity $G_{513}$. (c) Spectra of the fluorescence capacity $G_\lambda$ within two height ranges, which are marked on plot (b) by magenta boxes. Results are presented for the temporal interval from 20:00 to 23:30 UTC on June 16, 2023.**

Vertical profiles of key particle properties such as the aerosol and fluorescence backscattering coefficients, $\beta_{355}$ and $B_\lambda$, along with the fluorescence capacity $G_{513}$ and the ratio $B_{560}/B_{438}$ are shown in Fig.2.    Within the PBL, where urban aerosol is dominant, the fluorescence capacity is relatively low, about $0.8\times10^{-6}$ nm$^{-1}$.  However, within the smoke layer $G_{513}$ shows a significant increase with height, starting from $4\times10^{-6}$ nm$^{-1}$ at 8 km and rising to $12\times10^{-6}$ nm$^{-1}$ at 10 km. As mentioned, the main contribution to the smoke fluorescence is provided by the organic fraction. Thus, the height dependece of $G_{513}$ within the

smoke layer may indicate changes in the relative content of the organic fraction: it is minimal near the layer bottom and maximal in the center.

The spectral dependence of the fluorescence capacity for three height intervals is depicted in Fig.2c. Within the PBL, the fluorescence capacity of urban aerosol gradually decreases with wavelength. The second interval (5.5-6.5.km) emcompasses



a weak smoke layer in the FT. The fluorescence capacity within this layer increases for all wavelengths with a noticeable

maximum shifting to the 513 nm channel. The third interval (8-10 km) corresponds to the smoke layer in the UTLS. Here, the

maximum fluorescence shifts to the 560 nm channel and the fluorescence capacity at 560 nm increases by approximately a

factor 3 compared to the FT. This shift and increase in fluorescence capacity highlight the changes in the chemical composition

and concentration of organic compounds as the smoke ascends to higher altitudes.

From the spectra in Fig.2c, it is evident that urban aerosol and smoke can be efficiently separated by the fluorescence capacity.

The most efficient separation occurs in the 513 nm and 560 nm channels. The fluorescence spectra also offer a distinctive way

to differentiate between smoke and urban aerosols. The ratio of the fluorescence capacities, such as $\frac{B_{560}}{B_{438}}$ , is particularly useful

for distinguishing these two aerosol types. As depicted in Fig.2a, this ratio is approximately 0.4 within the PBL, indicating a

dominance of urban aerosol. It increases to about 1.6 within the smoke layer in the UTLS, reflecting the higher fluorescence

capacity and organic content of smoke at higher altitudes.

As discussed in Section 2, the smoke volume and mass concentrations can be estimated from the aerosol backscattering

coefficient using the conversion factors. The mean value of $\beta_{355}$ within the 8-10 km interval is 0.62 Mm$^{-1}$sr$^{-1}$. Thus, the volume

and mass concentrations within the UTLS layer are estimated to be 2.4±0.6 µm$^3$cm$^{-3}$ and 2.7±0.7 µg/m$^3$ respectively.

For comparison, Fig.3 shows the results of smoke observations in the FT on 16 June 2023. During this period, a strong smoke

layer with a fluorescence backscattering coefficient $B_{513}$ reaching up to 5 Tm$^{-1}$sr$^{-1}$nm$^{-1}$ extends from 5.0 to 8.5 km throughout

the night. The vertical profiles are depicted in Fig.4, highlighting the same particle properties as in Fig.2. Within the PBL the

fluorescence capacity $G_{513}$ is similar to the values observed on September 6-7, about 0.8×10$^{-6}$ nm$^{-1}$. Within the smoke layer,

$G_{513}$ increases up to 4×10$^{-6}$ nm$^{-1}$. However, unlike the results in Fig.2b, $G_{513}$ does not show significant height variations,

suggesting that the smoke composition within the layer remains consistent.

From the results presented in this section, we conclude that the fluorescence capacity of smoke at the UTLS is higher than at

the FT, and the maxima of fluorescence shifts from the 513 nm channel to the 560 nm channel. Additionally, smoke and urban

aerosol exhibit distinctly different spectral dependencies of $G_\lambda$. To corroborate these conclusion, we conducted an analysis of

numerous smoke episodes. The corresponding results are presented in the next section.

## 3.2 Smoke observations during the May-September 2023 period over Moscow

Smoke transported over the Atlantic was regularly observed over Moscow during May-September 2023, within both the FT

and UTLS. The results of these smoke measurements are summarized in Fig.5, which displays the height of the smoke layer's

center, $H_{sm}$, along with the smoke backscattering coefficient, $\beta_{355}$, and the fluorescence capacities $G_{513}$ and $G_{560}$ averaged over

the layer. The values observed within the FT and the UTLS are represented by open and solid symbols, respectively. The

highest smoke backscattering coefficient in the UTLS, $\beta_{355}$=3.2 Mm$^{-1}$sr$^{-1}$, was observed on 20 June at a height of 10.5 km, with

the corresponding smoke mass concentration in the layer estimated to be 14 µg/m$^3$.





Fig.5b shows that the fluorescence capacities $G_{513}$ and $G_{560}$ are generally higher in the UTLS than in the FT. Additionally, $G_{560}$

typically exceeds $G_{513}$ in the UTLS. Another important observation is that the fluorescence capacity exhibits a wide range of

values. Specifically, for $G_{560}$ in the UTLS this range is between $(2.5\text{-}11.0)\times10^{-6}$ nm$^{-1}$. The variations are likely related to

changes in the relative concentration and chemical properties of organic compounds.

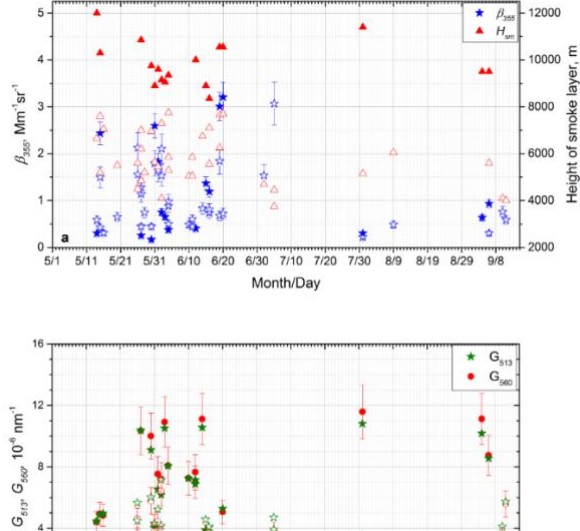

**Figure 5: Smoke layers in the FT (open symbols) and UTLS (solid symbols) over Moscow during May-September 2023. (a) The**
**height of the center of the smoke layer, $H_{sm}$, and the mean aerosol backscattering coefficient, $\beta_{355}$, within the layer. (b) The**
**fluorescence capacities $G_{513}$ and $G_{560}$ of smoke.**

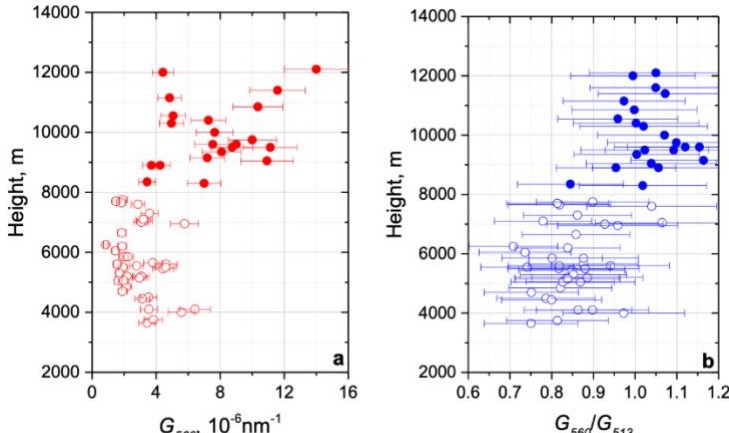

**Figure 6: Height dependence of (a) the fluorescence capacity $G_{560}$ and (b) the ratio $G_{560}/G_{513}$ for the smoke episodes during 2023.**
**Open and solid symbols correspond to the layers at the FT and UTLS respectively.**



To analyze the height dependence of the smoke fluorescence-related properties, Fig.6 provides the fluorescence capacity $G_{560}$ and the ratio $G_{560}/G_{513}$ for the smoke episodes observed in 2023 as a function of height. The data reveal that while the fluorescence capacity does not exhibit a clear trend within the FT, it increases above 8 km. Despite considerable variability in the data points, the mean value of $G_{560}$ at the UTLS is 2.7 times higher than in the FT. Additionally, the mean ratio $G_{560}/G_{513}$ is 0.85 in the FT and 1.03 in the UTLS, indicating a tendency for the fluorescence maximum to shift of toward the 560 nm channel with height.

The systematic difference between the fluorescence spectra of urban particles and smoke was consistently observed during the wildfire season. Fig.7 shows the spectra of $B_\lambda$ and $G_\lambda$ recorded in 2023. The data are presented for three height intervals, in the PBL, FT and in the UTLS. To compare the "shape" of the fluorescence spectra for different days, $B_\lambda$ values are normalized to the sum $\sum_{i=1}^{5} B_{\lambda_i}$ .

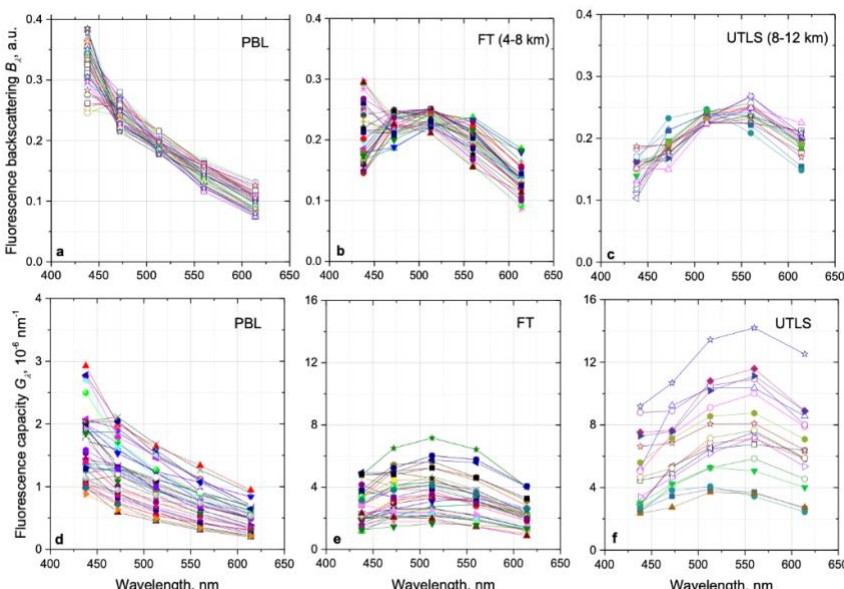

**Figure 7: Spectra of (a-c) the fluorescence backscattering, $B_\lambda$, and (b-d) the fluorescence capacity, $G_\lambda$, measured over Moscow during May-September, 2023. Results are given for three height intervals: the PBL, FT (4-8 km) and UTLS (8-12 km). The fluorescence backscattering coefficients are normalized to the sum $\sum_{i=1}^{5} B_{\lambda_i}$ .**





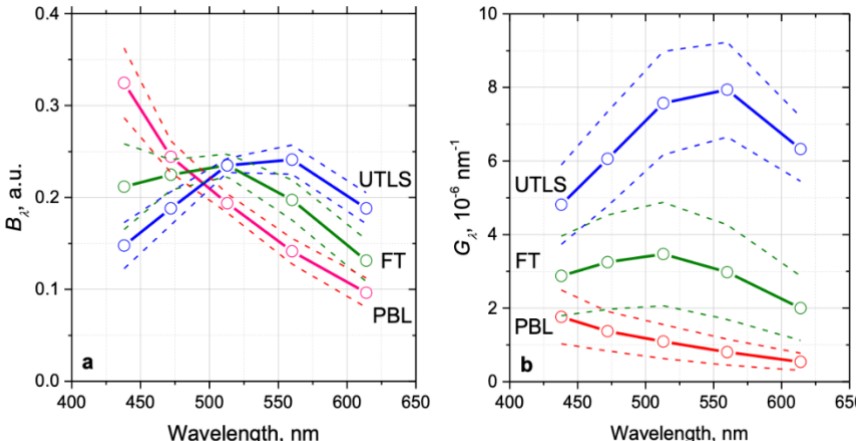

**Figure 8: Averaged values (solid lines) and standard deviations (dash lines) of (a) the fluorescence backscattering $B_\lambda$ and (b) the fluorescence capacity $G_\lambda$ measured over Moscow during May-September 2023. Results are provided for the PBL, FT and UTLS.**


Fig. 8a depicts the mean values of normalized $B_\lambda$ along with their standard deviations. Within the PBL, where the urban aerosols prevail, normalized $B_\lambda$ shows significant deviations from the mean value, particularly around 10% at the 438 nm channel. Despite these deviations, all measurement sessions reveal that the fluorescence backscattering gradually decreases with wavelength. In the smoke layers at the FT, the fluorescence backscattering exhibits a maximum at the 513 nm channel.

The highest deviations are observed at the 438 nm channel, around 20%, likely due to the mixing of smoke with urban particles during transport. At the UTLS, the fluorescence backscattering peaks at the 560 nm channel, with deviations from the mean value being lower, approximately 15% at the 438 nm channel.

The variability of $G_\lambda$ is notably greater compared to normalized $B_\lambda$, as shown in Fig.7d-f. Variations of $G_\lambda$ within the PBL could be influenced by the hygroscopic growth of urban aerosol. Additionally, the fluorescence capacity could increase due to

the presence of smoke at the PBL. In the FT, variation in $G_\lambda$ at the FT might be due to mixing with urban aerosol during transport and the mixing with descending smoke layers from the UTLS. Even at the UTLS, where such mixing effects are minimal, $G_{513}$ ranges within $(4.0\text{-}13.5)\times10^{-6}$ nm$^{-1}$. This variability is likely related to changes in the smoke's chemical composition, particularly the organic fraction. Despite these variations, the mean values of fluorescence capacity for all channels, as shown in Fig.8b, exhibit a clear increase with height. Specifically, $G_{513}$ is approximately $1.0\times10^{-6}$ nm$^{-1}$ in the PBL,

$3.5\times10^{-6}$ nm$^{-1}$ in the FT, and $7.6\times10^{-6}$ nm$^{-1}$ in the UTLS.




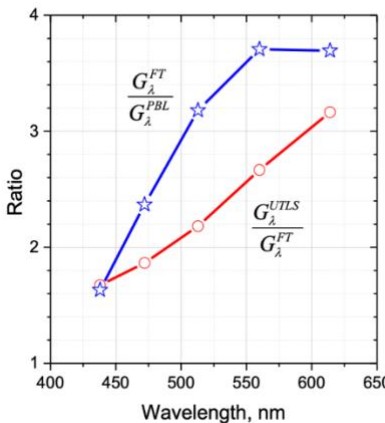

**Figure 9: The spectral dependence of the ratio of the averaged $G_\lambda$ from Fig.9b in the FT to that in the PBL, $\frac{G_\lambda^{FT}}{G_\lambda^{PBL}}$ , along with the corresponding ratio for the UTLS and FT, $\frac{G_\lambda^{UTLS}}{G_\lambda^{FT}}$ .**

In Fig. 9, the spectral dependence of the ratio of the mean fluorescence capacity in the FT to that in the PBL, $\frac{G_\lambda^{FT}}{G_\lambda^{PBL}}$ . is shown.

This ratio increases with wavelength, reaching a maximum value of 3.7 at the 560 nm channel. Similarly, the ratio of the mean fluorescence capacity in the UTLS to that in the FT, $\frac{G_\lambda^{UTLS}}{G_\lambda^{FT}}$ , also increases with wavelength, with the highest value of 3.2 observed at the 614 nm channel. The observed ratio $\frac{G_\lambda^{UTLS}}{G_\lambda^{FT}}$ suggests, that the concentration of organic compounds, which have a fluorescence spectrum shifted towards longer wavelengths, is higher in the UTLS compared to the FT.

The separation of urban and smoke particles, as discussed in Veselovskii et al. (2022b, 2024) has used a "threshold algorithm" based on fluorescence capacities at 466 nm to classify particles as smoke if their fluorescence capacity exceeds a certain threshold. While effective, this approach is sensitive to the range of $G_\lambda$ variations, which, as follows from Fig.8, can be considerable. Additionally, the hygroscopic growth of particles reduces fluorescence capacity, potentially impacting classification accuracy. An alternative to this method is the "spectral" approach, which utilizes the distinct fluorescence spectra of smoke and urban particles. This method may offer an advantage, particularly in varying environmental conditions such as changes in relative humidity.





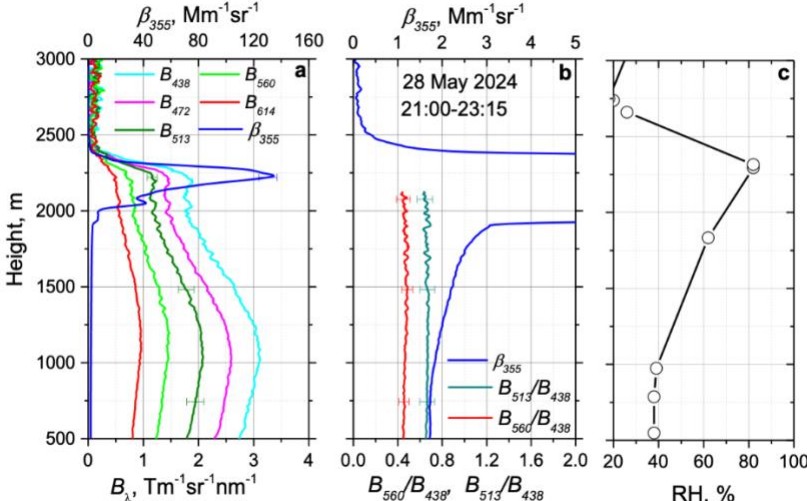

**Figure 10: Fluorescence measurements in the presence of aerosol hygroscopic growth on May 28, 2024. (a) Vertical profiles of the fluorescence backscattering coefficients $B_\lambda$ at the wavelengths 438, 472, 513, 560, and 614 nm along with the aerosol backscattering coefficient $\beta_{355}$. (b) Scaled $\beta_{355}$ and the ratios $B_{513}/B_{438}$, $B_{560}/B_{438}$. (c) Profile of the RH measured by a radiosonde on 29 May at 00:00 UTC.**

The fluorescence spectrum is not influenced by the RH. This is supported by Fig.10, which displays the aerosol and the fluorescence backscattering coefficients ($\beta_{355}$ and $B_\lambda$) measured at the conditions of particle hygroscopic growth on May 28, 2024. The RH, as recorded by the radiosonde in Dolgoprudny, increased with height from 40% at 1000 m to 80% at 2250 m, along with a corresponding rise in $\beta_{355}$. Despite these changes in RH and the presence of hygroscopic growth, the ratio of fluorescence backscattering, such as $B_{560}/B_{438}$ and $B_{560}/B_{438}$ (Fig.10b) remains constant, even within the cloud, indicating that the fluorescence spectrum is not affected by changes in RH.

**3.3 Distinguishing the fluorescence contributions of smoke and urban particles.**

Smoke and urban aerosol exhibit distinct spectral dependence for $B_\lambda$, and the averaged normalized $B_\lambda$ shown in Fig.8a can be taken as the reference spectra. Thus, if smoke and urban particles are predominant components of an aerosol mixture, the observed fluorescence spectrum can be approximated as a combination of their contributions ($B_\lambda^U$ and $B_\lambda^S$)

$$B_\lambda = B_\lambda^U + B_\lambda^S = aB_\lambda^{Uref} + bB_\lambda^{Sref} \tag{6}$$

Here $B_\lambda^{Uref}$ and $B_\lambda^{Sref}$ are the reference fluorescence spectra of urban particles and smoke, respectively. Given the five fluorescence channels, the system (6) contains five equations with two unknowns. This system was solved by the least squares method to derive $a$ and $b$ coefficients. For the urban particles, the fluorescence spectrum within the PBL in Fig.8a was used as reference. For smoke, two reference spectra were considered, one for the FT and the other for the UTLS. The values $B_\lambda/B_{438}$,





for the reference spectra, are summarized in Table 1. With these values, the derived $B_\lambda^U$ and $B_\lambda^S$ can be recalculated from one wavelength to another. Additionally, from Table 1 one can conclude, that the ratio $B_{560}/B_{438}$ is a convenient parameter for distinguishing between smoke and urban particles. For the reference spectra this ratio is of 0.44 for the urban aerosol in the
PBL and 0.93 and 1.63 for smoke in the FT and UTLS respectively.

**Table 1. The reference spectra of the fluorescence backscattering coefficient for urban aerosol in the PBL, and for smoke in the FT and in UTLS. The values $B_\lambda$ are normalized to $B_{438}$.**

|  | $B_{438}$ | $B_{472}$ | $B_{513}$ | $B_{560}$ | $B_{614}$ |
|---|---|---|---|---|---|
| Urban (PBL) | 1 | 0.75 | 0.60 | 0.44 | 0.30 |
| Smoke (FT) | 1 | 1.06 | 1.11 | 0.93 | 0.62 |
| Smoke (UTLS) | 1 | 1.28 | 1.59 | 1.63 | 1.28 |


During the wildfire season of 2023, numerous episodes were observed where tropospheric smoke descended to the PBL and mixed with urban aerosol. Below, we apply the spectral approach to analyze two such episodes on September 25-26 and 26-27, 2023.

### 3.3.1 Case study: September 25-26, 2023. Smoke at the top of the PBL

On the night of September 25-26, 2023, the air masses above 2000 m over the observation site were transported across the Atlantic from the regions affected by North American wildfires, as indicated by the HYSPLIT Backward Trajectory Analysis (Stein et al., 2015) shown in Fig.11. The spatiotemporal distributions of key aerosol parameters, such as the aerosol and fluorescence backscattering coefficients ($\beta_{355}$, and $B_{513}$), fluorescence capacity ($G_{513}$), and the ratio $B_{560}/B_{438}$ are depicted in Fig.12. These parameters help distinguish between two different aerosol layers. In the first layer, extending from the surface
to approximately 1500 m altitude, the fluorescence capacity $G_{513}$ is low (about $1\times10^{-6}$ nm$^{-1}$) and the ratio $B_{560}/B_{438}$ is approximately 0.4. Both these characteristics indicate that the aerosol in this layer is of urban origin. Above 1500 m both the fluorescence capacity and the ratio $B_{560}/B_{438}$ increase, with $G_{513}$ exceeding $6\times10^{-6}$ nm$^{-1}$ and $B_{560}/B_{438}$ reaching about 1.6. These characteristics suggest the predominance of smoke particles. The radiosonde measurements reveal a temperature inversion at 1500 m, indicating that the lower layer is the PBL, while the smoke is located at its top.



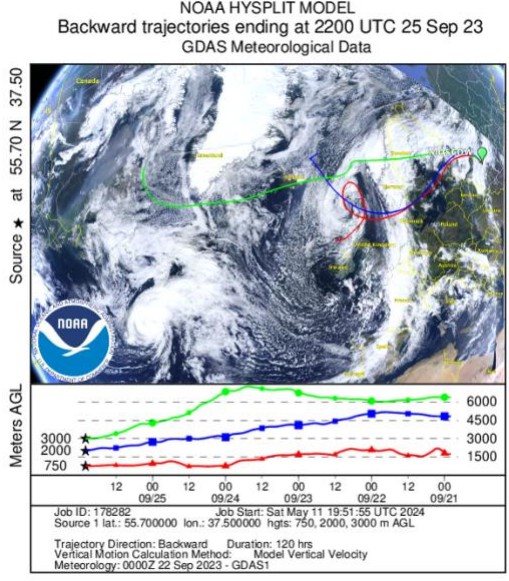

**Figure 11: The HYSPLIT five-day backward trajectories for the air mass over Moscow at altitudes 750 m, 2000 m, and 3000 m on September 25, 2023 at 22:00 UTC.**

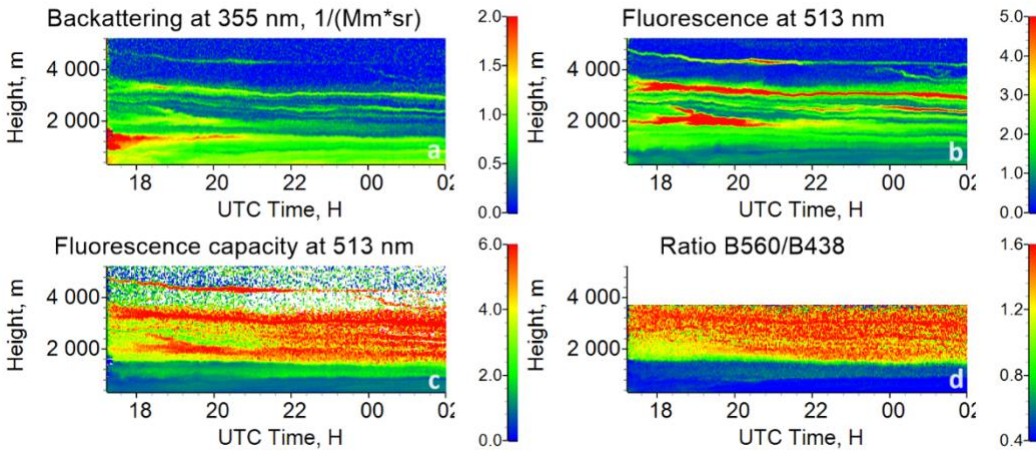

**Figure 12: Spatio-temporal distributions of (a) the aerosol backscattering coefficient $\beta_{355}$, (b) the fluorescence backscattering coefficient $B_{513}$ (in Tm$^{-1}$sr$^{-1}$nm$^{-1}$), (c) the fluorescence capacity $G_{513}$ (in $10^{-6}$ nm$^{-1}$), and (d) the ratio $B_{560}/B_{438}$ during the night of September 25-26, 2023.**





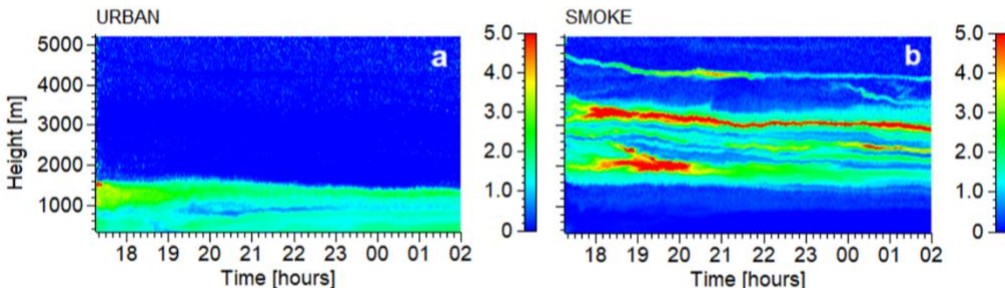

**Figure 13: The fluorescence backscattering coefficient (in Tm$^{-1}$sr$^{-1}$nm$^{-1}$) at 438 nm attributed to (a) urban particles and (b) smoke on September 25-26, 2023. The computations were performed using the smoke reference fluorescence spectrum in the FT.**

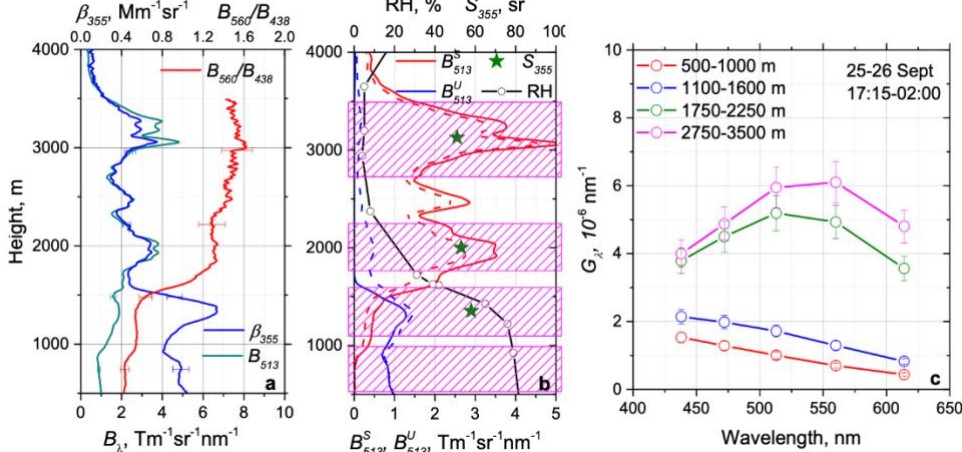

**Figure 14:  (a) Vertical profiles of the aerosol and fluorescence backscattering coefficients ($\beta_{355}$, $B_{513}$), along with the ratio $B_{560}/B_{438}$. (b) Contribution of smoke, $B_{513}^{S}$, and urban, $B_{513}^{U}$, particles to the overall fluorescence backscattering coefficient, $B_{513}$, alongside the relative humidity measured by a radiosonde and averaged lidar ratios, $S_{355}$. $B_{513}^{S}$ and $B_{513}^{U}$ were calculated using the reference spectra for smoke in the FT (solid lines) and UTLS (dashed lines). (c) Spectra of the fluorescence capacity, $G_\lambda$, within four height ranges. These ranges are highlighted in plot (b) with magenta boxes.**





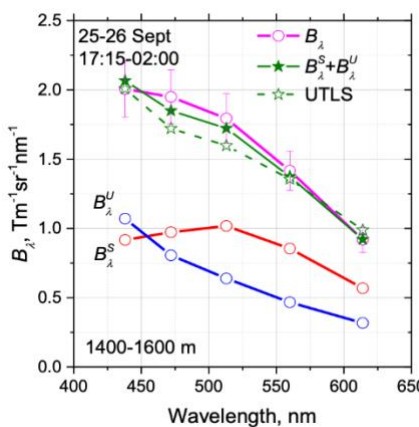

**Figure 15: The spectrum of the fluorescence backscattering coefficient, $B_\lambda$, within the 1400-1600 m interval along with the contributions of smoke, $B_\lambda^S$, urban aerosol, $B_\lambda^U$, and their sum, $B_\lambda^S + B_\lambda^U$. The computations were performed using the smoke reference fluorescence spectra in the FT (solid line) and in the UTLS (dashed line).**

The spectral approach described was applied to this case to distinguish the smoke and urban contributions to the overall fluorescence backscattering coefficient. The spatiotemporal distributions of $B_{438}^U$ and $B_{438}^S$ obtained for FT reference spectra are shown in Fig.13. Smoke and urban particles are well separated, with their mixing occurring only within a relatively thin layer (several hundred meters). Vertical profiles of particle parameters averaged over the interval from 17:15 to 02:15 UTC are displayed in Fig.14. The aerosol backscattering coefficient $\beta_{355}$ is the strongest within the PBL, while $\beta_{355}$ within the smoke layer (above 1500 m) is approximately a factor of 2 lower. The ratio $B_{560}/B_{438}$ rises with height from 0.4 to 1.6 due to an increase in the smoke contribution to the overall fluorescence backscattering. The profiles of $B_{513}^S$ and $B_{513}^U$ are depicted in Fig.14b. Below 1000 m, urban particles dominate, while smoke becomes prevalent above 1600 m. Within the 1000-1600 m range, the particles are mixed, resulting in a change in the fluorescence spectrum with height. To understand the sensitivity of the results to the choice of the reference spectrum, calculations of $B_{513}^S$ and $B_{513}^U$ in Fig.14b was performed using the smoke reference spectrum for the FT (solid lines) and for the UTLS (dashed lines). The results obtained are similar, with the maximal difference between $B_{513}^S$ obtained with these two reference spectra occurring at 2000 m, where it is below 25%.

Fig.14b provides also the averaged lidar ratios at 355 nm, $S_{355}$, within the highlighted ranges. The values of $S_{355}$ for two height intervals within the smoke layer are about 50 sr, which agrees with the range of $S_{355}$ variation (20-50 sr), reported by Ansmann et al. (2021) and Hu et al. (2022) for aged North-American smoke. Within 1100-1600 m layer $S_{355}$ is higher (about 60 sr) due to the presence of urban particles.

The spectra of the fluorescence capacity for four height ranges are depicted in Fig.14c. Below 1000 m, the spectrum exhibits characteristics typical of urban particles: $G_\lambda$ decreases with wavelength and $G_{513}$ is about $1.0 \times 10^{-6}$ nm$^{-1}$, which is close to the



corresponding value in Fig.8b. Within 1100-1600 m height range, $G_{513}$ increases up to $1.8 \times 10^{-6}$ nm$^{-1}$ due to the presence of smoke. The maxima of the fluorescence spectrum of smoke within the 1750-2250 m layer occurs at 513 nm channel, and for the 2750-3500 m range, this maximum shifts to 560 nm. The value of $G_{513}$ is high (up to $6.0 \times 10^{-6}$ nm$^{-1}$), indicating that the

layer at 3000 m may contain the smoke descended from the upper troposphere, which agrees with results of the BT analysis in Fig.11.

To test how well the calculated spectrum of the fluorescence backscattering ( $B_\lambda^S + B_\lambda^U$ ) matches the observations ($B_\lambda$) Fig.15 depicts the corresponding spectra for 1400-1600 m height interval, where urban particles are mixed with smoke. Computations were performed using the smoke reference spectra for the FT and the UTLS. The spectrum for the FT provides a better match

to observations, suggesting that its use is probably more aproppriate.

### 3.3.2 Mixing of smoke and urban particles within the PBL on September 26-27, 2023

On the night September 26-27, 2023, the air masses transported from North America descended from approximately 7000 m to 2000 m, as indicated by the BT analysis in Fig.16. Thus, smoke could be mixed with urban particles within the PBL. The spatiotemporal distributions of the particle parameters, such as $\beta_{355}$, $B_{513}$, $G_{513}$, and $B_{560}/B_{438}$ are shown in Fig.17. The aerosol

backscattering decreases fast above 1200 m, and the radiosonde measurements reveal a temperature inversion at the same height. Similar to the observations on 25 September (Fig.12), the fluorescence capacity and the ratio $B_{560}/B_{438}$ below 1200 m are lower compared to the values above the PBL. From distributions in Fig.17, one can conclude that on September 26-27, as well as on the previous night, the smoke is located at the top of the PBL. However, the spatiotemporal distributions of $B_{438}^U$

and $B_{438}^S$ in Fig.18 reveal a significant smoke fraction within the PBL.

Vertical profiles of particle properties averaged over the time interval ranging from 17:10 to 23:00 UTC are displayed in Fig.19. The RH measured by the radiosonde increases with height, reaching 80% at 1400 m. The increase of $\beta_{355}$ near the top of the PBL is likely related to particle hygroscopic growth. The ratio $B_{560}/B_{438}$ is about 0.65 within the PBL, which is higher than on previous night and may be due to the presence of smoke. Above the PBL, where smoke is predominant, this ratio increases to approximately 1.4. The profiles of $B_{513}^S$ and $B_{513}^U$ are calculated using the reference spectra of smoke both in the

FT (solid line) and UTLS (dashed line). The choice of the reference spectrum influences the results. In particular, the smoke contribution $B_{513}^S$ within the PBL obtained with the reference spectra for the FT and UTLS are of 0.9 Tm$^{-1}$sr$^{-1}$nm$^{-1}$ and 0.6 Tm$^{-1}$sr$^{-1}$nm$^{-1}$ respectively.



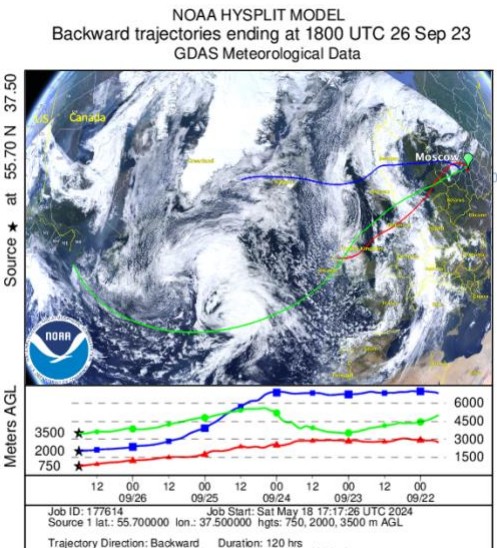

**Figure 16: The HYSPLIT five-day backward trajectories for the air mass over Moscow at altitudes 750 m, 2000 m, and 3500 m on September 26, 2023 at 18:00 UTC.**

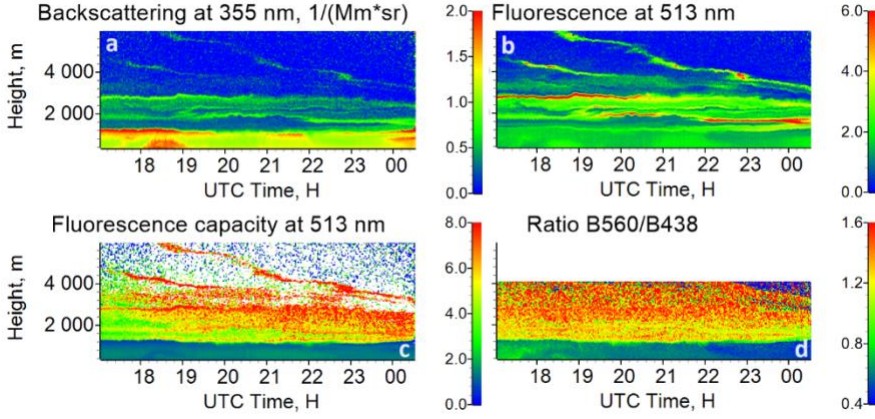

**Figure 17: Spatio-temporal distributions of (a) the aerosol backscattering coefficient $\beta_{355}$, (b) the fluorescence backscattering coefficient $B_{513}$ (in Tm$^{-1}$sr$^{-1}$nm$^{-1}$), (c) the fluorescence capacity $G_{513}$ (in 10$^{-6}$ nm$^{-1}$), and (d) the ratio $B_{560}/B_{438}$ during the night of September 26-27, 2023.**



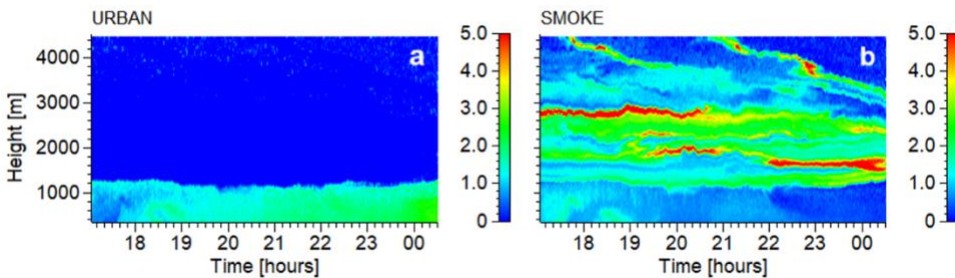

**Figure 18: The fluorescence backscattering coefficient (in Tm$^{-1}$sr$^{-1}$nm$^{-1}$) at 438 nm wavelength attributed to (a) urban particles ( $B_{438}^{U}$ ) and (b) smoke ( $B_{438}^{S}$ ) on September 26-27, 2023. The computations were performed using the smoke reference fluorescence spectrum in the FT.**

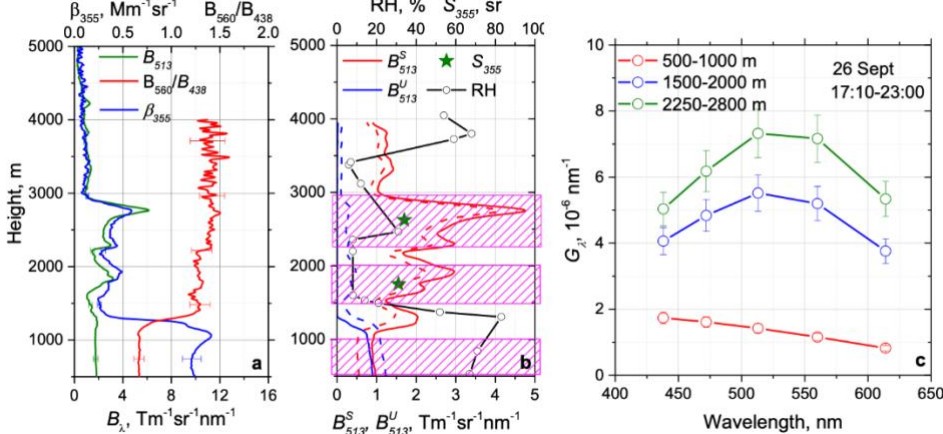

**Figure 19: (a) Vertical profiles of the aerosol and fluorescence backscattering coefficients ($\beta_{355}$, $B_{513}$), along with the ratio $B_{560}/B_{438}$. (b) Contribution of smoke, $B_{513}^{S}$, and urban, $B_{513}^{U}$, particles to the overall fluorescence backscattering coefficient, $B_{513}$, alongside the relative humidity measured by a radiosonde and averaged lidar ratios, $S_{355}$. $B_{513}^{S}$ and $B_{513}^{U}$ were calculated using the reference spectra for smoke in the FT (solid lines) and UTLS (dashed lines). (c) Spectra of the fluorescence capacity, $G_\lambda$, within four height ranges. These ranges are highlighted in plot (b) with magenta boxes.**

The spectra of the fluorescence capacity are shown in Fig.19c for three height ranges. Within the PBL, $G_\lambda$ decreases with wavelength, but the value $G_{513}$=1.4×10$^{-6}$ nm$^{-1}$ is higher than on the previous night, due to the contribution of smoke. Above the PBL the maximum of the spectrum occurs at 513 nm, with $G_{513} \approx 7.2×10^{-6}$ nm$^{-1}$ for the 2250-2800 m height interval. Such high fluorescence capacity is typically observed in the UTLS, supporting the conclusion from the BT analysis that the smoke at this height originates from the upper troposphere.



As discussed in the Sect.2, the smoke mass concentration within the PBL can be estimated if the smoke contribution $\beta_{355}^{S}$ to

the total aerosol backscattering $\beta_{355}$ is known. This contribution can be calculated as $\beta_{355}^{S} = \dfrac{B_{513}^{S}}{G_{513}}$. The fluorescence capacity

of smoke varies widely, but assuming it remains constant within the 500-2000 m interval, we can use the value at 2000 m $G_{513}$

$\approx 5.5 \times 10^{-6}$ nm$^{-1}$. The smoke contribution $\beta_{355}^{S}$ to the aerosol backscattering coefficient within the PBL, calculated using the

reference fluorescence spectra for the FT and UTLS is of 0.16 Mm$^{-1}$sr$^{-1}$ and 0.11 Mm$^{-1}$sr$^{-1}$ respectively.

An estimation of $\beta_{355}^{S}$ can also be made based on the fluorescence capacity of smoke and urban aerosol. Using Eq.5 and the

$G_{\lambda}^{S}$ values from Fig.20c within 1500-2000 m interval for different fluorescence channels, we obtain $\beta_{355}^{S}$ varying within the

0.09-0.11 Mm$^{-1}$sr$^{-1}$ range. The values obtained are lower than those derived using the spectral approach. Nevertheless, the

results obtained with the two different approaches are fairly consistent, considering the numerous assumptions made. Finally,

the mass concentration of smoke within the PBL is estimated to be in the 0.4-0.7 µg/m$^3$ range.

**4 Conclusion**

Regular observations conducted during the wildfire season using the five-channel fluorescence lidar enabled the detection of

spectral variations in fluorescence capacity across a wide range of altitudes, from the planetary boundary layer (PBL) to the

upper troposphere-lower stratosphere (UTLS). From results obtained we can conclude that the fluorescence capacity of smoke

varies significantly from episode to episode. Notably, in the UTLS, where $G_{\lambda}$ is not influenced by urban aerosol, the $G_{560}$

varies within the range of $(4-14) \times 10^{-6}$ nm$^{-1}$. This variability suggests substantial differences in the composition of the smoke

observed during different episodes. In contrast, the shape of the fluorescence spectrum, represented by the normalized

fluorescence backscattering coefficient does not exhibit such strong variations. Given that the primary contribution to

fluorescence comes from the organic fraction of smoke, it is reasonable to assume that the observed variations in $G_{\lambda}$ are due

to changes in the relative concentration of organic components. The factor of more than three change in $G_{\lambda}$ suggests that the

proportion of organic materials in the smoke particles can vary significantly, reflecting differences in the sources, combustion

conditions, or aging processes of the smoke.

The analysis of our unique dataset also reveals a notable height dependence of the fluorescence capacity. Despite strong

variations, there is a consistent tendency for the fluorescence capacity at all wavelengths to increase with height. Specifically,

mean value of $G_{513}$ at the UTLS is approximately twice as high as that in the FT. This trend may indicate an increase in the

organic fraction of smoke particles with altitude. Moreover, the height dependence of $G_{\lambda}$ intensifies with increasing

wavelength. The most significant change in fluorescence capacity between the UTLS and FT is observed in the 614 nm

channel. Consequently, the peak of the fluorescence spectrum shifts towards longer wavelengths as altitude increases. This



spectral shift suggests that the chemical composition of the smoke, particularly the organic components, evolves with altitude, potentially due to differential aging processes.

The results accumulated throughout 2023 revealed a clear distinction between the fluorescence spectra of smoke and urban aerosol. The fluorescence spectrum of urban aerosol, predominantly located within the PBL, consistently decreases with wavelength. In contrast, the fluorescence of smoke exhibits a peak that shifts toward the 513 nm or 560 nm channel. Notably, the fluorescence capacity of smoke is higher across all wavelengths compared to urban aerosol, with the most pronounced differences observed at the 513 nm and 560 nm channels. This difference can be effectively utilized to differentiate between
urban and smoke fractions in an aerosol mixture.

To discriminate between smoke and urban particles based on their fluorescence spectra, at least two fluorescence channels are necessary. Based on our dataset, the 438 nm and 560 nm channels would provide the most efficient separation. However, if modifications are made to a standard Mie-Raman lidar, preserving the capability for 3β+2α observations could be advantageous. In this case, incorporating the 438 nm and 513 nm channels could be a practical solution.


*Data availability*. Lidar measurements are available upon request

(igorv@pic.troitsk.ru).

*Author contributions*. IV processed the data and wrote the paper. MK and BB prepared the program for aerosol mixture
partitioning. NK performed the measurements. QH, PG and TP performed data analysis and helped with manuscript preparation.

*Competing interests*. The authors declare that they have no conflict of interests.

**Acknowledgement.** We acknowledge CaPPA project (ANR-11-LABX-0005-01) for funding observation-related scientific
activities and OBS4CLIM project (ANR-21-ESRE-0013) for providing financial support to Q. Hu. The Russian Science Foundation is acknowledged for its supports in the frame of project 21-17-00114

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
