# Peer review of "Fluorescence properties of long-range transported smoke: Insights from five-channel lidar observations over Moscow during the 2023 wildfire season"

_EGUsphere, 2024_

## Referee Comment (RC2)

egusphere-2024-2874 | Journal relation: ACP    Submitted on 13 Sep 2024
**Fluorescence properties of long-range transported smoke: Insights from five-channel lidar observations over Moscow during the 2023 wildfire season**
*Igor Veselovskii, Mikhail Korenskiy, Nikita Kasianik, Boris Barchunov, Qiaoyun Hu, Philippe Goloub, and Thierry Podvin*

This manuscript is very well written, with the quality of the text being one of the positive aspects to highlight. The analysis carried out deepen the study of the aerosol biomass burning (BBA) transported over long distances including entrainment in the PBL. Based on the studies that highlight the advantage of measuring the total fluorescence spectrum, the authors use several discrete channels and obtain the spectral properties using the width of the transmission band. By using five different wavelengths, they can evaluate their performance. According to the results obtained, a current Raman lidar equipment can be significantly improved for aerosol typing by including only two fluorescence channels. This greatly contributes to current knowledge and allows using the contrasted data for future research and further advancing the characterization of atmospheric particles.

On the other hand, something that made me think was that the analysis was only carried out in the PBL, FT and UTLS. Furthermore, considering 8 to 12 km as UTLS. Perhaps 8km would not be considered as upper troposphere and 12km as low stratosphere either. This is something that should be considered and more appropriate language should be used or why it is treated this way should be explained.

Additionally, there are several points that could be interesting to include with the aim to have a more consistent work.

As it happens, it could be included the analysis of the particle depolarization ratio once smoke and urban particles are classified. Be that as it may, something that undoubtedly devalues the work is the quality of the figures. Its size and the size of the axes and legends, the half-titles in some cases or the lack of units in the figure itself (Figure 1 right), the letter that differentiates one subfigure from another such as a, b, c cannot be seen (Figure 2), the height range of the graphs of the same event differ from each other (Figures 5 and 6), the meaning of the different colors of the data series is not explained (Figure 7).

Another issue to point out is the fact that many are times that average period to study and plot is not indicated.

Some specific details can be found below:

Page 1 line 16

How long are the 40 smoke episodes? Are they all equally representative? How many hours of measurement does each of them include? The lidar measures 24/7?

Page 2 line 45-48

It would be advisable to explain in detail that hygroscopic growth decreases the fluorescence capacity but does not affect the fluorescence spectrum.

Page 10 line 23

The text points that despite standard deviations, all measurement sessions reveal that the fluorescence backscattering gradually decreases with wavelength but Figure 8a fails to show that.

Page 14 Figure 12

The word "Backscattering" in the upper left subfigure should be corrected.

Page 16 line 322

Perhaps the average interval established to obtain the properties of aerosol particles is too long in time considering the extent and detail of the analysis of the events.

In summary, this paper represents a major advance in Raman and fluorescence-based lidar techniques, but the presentation of the work needs to be revised to maintain the high quality of the scientific content.

---

## Author Comment (AC1)

**Response to Referee #1**

First, we are very grateful to the Referee for accurate reading the manuscript and useful suggestions.

*The paper is well written and deals with an important aerosol fluorescence lidar application. The authors show how spectrally resolved aerosol fluorescence information can be used to distinguish local boundary layer aerosol and smoke in the free troposphere (after long-range transport). I have only minor comments:*

*Page 1, line 27, smoke influence on cirrus clouds is nicely shown by Mamouri et al., ACP, 2023. Should be cited!*

Done

*Page 2, line 46 and 48: It is confusing that the fluorescence capacity depends on hygroscopic growth, and the fluorescence spectrum does not…. Please explicitly explain that this is related to the humidity dependence of the total backscatter coefficient.*

Corresponding sentence is added.

*Figure 2: larger letters are needed for the different panels (a), (b), (c)….. this holds for most figures.*

Fonts are increased

*The separation or division of the altitude range in PBL, free troposphere and UTLS is not satisfactory. One should better use words 'lower free troposphere' and maybe 'middle and upper troposphere'. The UTLS defines the region around the tropopause and frequently suggests interaction between the upper troposphere and lower stratosphere. However, you discuss and distinguish smoke in the lower to middle free troposphere and smoke in the middle to upper troposphere. So please state that accordingly. Avoid UTLS if you discuss smoke in the upper troposphere only.*

We agree with reviewer. In the revised manuscript we use "Lower Free Troposphere (LFT) and "Middle and Upper Troposphere (MUT)".

*Section 3.2.*
*Figure 5, …. (a) and (b) are too small, the symbols are too small, the legend is too small, everything is not easy to read (in the printout) and thus it is difficult to get the main message.*

Figures are modified

*By studying the figures, the question arises: Why is that? What are the reasons for the differences? Maybe the higher smoke layers are lofted by convective cloud activity, and the lower layers are just emitted and ascend as a result of sunlight absorption. Aging of smoke depends on the availability of gases (emitted together with the particles) and is faster when the humidity is high….. The probability for faster aging is higher in the lower free troposphere than in the typically dry upper free troposphere. With time and condensation of gases the BC fraction decreases typically towards a few percent, and at the same time the organic carbon fraction increases…. Can these arguments explain your observations better? At least as a reader I missed reasons for the findings*

*...dynamical aspects, injection aspects, transport aspects, chemical aging, cloud processing and aging…. Even if only hypotheses can be presented, this will stimulate further research.*

These questions are indeed crucial, and our study offers valuable new insights. It is now well recognized that the chemical aging of smoke increases the organic carbon (OC) fraction, thereby enhancing fluorescence capacity. Our measurements reveal that fluorescence capacity—and thus the OC fraction—tends to increase with altitude. One possible explanation is that smoke aging in the upper troposphere may occur more rapidly, despite lower relative humidity. This aging process is influenced by various factors, including injection parameters and transport conditions. However, at this stage, we are unable to identify the exact mechanisms involved, and further research is needed to clarify these dynamics. Corresponding comment is added to the manuscript.

*Figure 6 seems to support that different vertical transport phenomena are active for smoke below and above 8 km height above Moscow.*

This is probably true not only for Moscow. Results obtained in Lille (manuscript is in preparation) demonstrate that smoke particles size increases with height. Thus, smoke particles in lower and upper troposphere have not only different fluorescence properties, but the microphysical parameters as well.

*Figure 7: again nice results, clear differences, and the question arises? What controls these differences? Again, (a), (b), (c)…. too small. And different scales in (d) vs (e) and (f) is not helpful.*

Figures are modified

*Section 3.3 should be shortened. The article is quite long, and two extended case studies in Section 3.3 are probably too much for the reader.*

In revised manuscript we shortened Section 3.3. Second case is removed.

*I missed the depolarization ratio observations. Was the particle depolarization ratio always close to zero? It may have been significantly enhanced in the dry upper troposphere.*

Unfortunately, at current configuration of the lidar, the depolarization ratio is not available.

*The conclusion section does not mention the results in section 3.3, maybe one should remove section 3.3?*

We shortened this section, but prefer to keep it, because it is important to demonstrate separation of smoke and urban particles based on their fluorescence spectra. Corresponding comment is added to Conclusion.

*Figures 11 and 16: Because of the clouds it is difficult to see the pathways of the smoke transport, over the continents and over the ocean.*

Yes, we agree. The figure is modified.

*Page 16, line 332: smoke lidar ratios at 355 nm of down to 20 sr? Values of 35-60 sr for 355 nm.*

In Ansmann et al (2021) variation of S355 is 35-50 sr, while Hu et al, (2022) report lower values (25-50 sr). Corresponding correction is introduced in manuscript.

*A final remark: The probability for pure smoke is highest in the upper troposphere and therefore the fluorescence spectrum for this smoke may be used as reference spectrum.*

From observations performed in Lille and Moscow we can conclude, that fluorescence properties of smoke in the lower and upper troposphere are different. As discussed above, these properties are influenced by the processes of smoke aging and transportation. So, by our opinion, we should use different smoke spectra for lower and upper troposphere. However, in our algorithm, the choice of the spectra does not influence significantly the results.

---

## Author Comment (AC2)

**Response to Referee #2**

First, we are very grateful to the Referee for accurate reading of manuscript and useful suggestions.

*This manuscript is very well written, with the quality of the text being one of the positive aspects to highlight. The analysis carried out deepen the study of the aerosol biomass burning (BBA) transported over long distances including entrainment in the PBL. Based on the studies that highlight the advantage of measuring the total fluorescence spectrum, the authors use several discrete channels and obtain the spectral properties using the width of the transmission band. By using five different wavelengths, they can evaluate their performance. According to the results obtained, a current Raman lidar equipment can be significantly improved for aerosol typing by including only two fluorescence channels. This greatly contributes to current knowledge and allows using the contrasted data for future research and further advancing the characterization of atmospheric particles.*

*On the other hand, something that made me think was that the analysis was only carried out in the PBL, FT and UTLS. Furthermore, considering 8 to 12 km as UTLS. Perhaps 8km would not be considered as upper troposphere and 12km as low stratosphere either. This is something that should be considered and more appropriate language should be used or why it is treated this way should be explained.*

Yes, similar comment was provided also by Referee 1. In the revised manuscript for these height ranges we use "Lower Free Troposphere (LFT) and "Middle and Upper Troposphere (MUT)".

*Additionally, there are several points that could be interesting to include with the aim to have a more consistent work. As it happens, it could be included the analysis of the particle depolarization ratio once smoke and urban particles are classified.*

We agree, that depolarization measurements are important. Unfortunately, at the current configuration of the lidar, these measurements are not available.

*Be that as it may, something that undoubtedly devalues the work is the quality of the figures. Its size and the size of the axes and legends, the half-titles in some cases or the lack of units in the figure itself (Figure 1 right), the letter that differentiates one subfigure from another such as a, b, c cannot be seen (Figure 2), the height range of the graphs of the same event differ from each other (Figures 5 and 6), the meaning of the different colours of the data series is not explained (Figure 7).*

In the revised manuscript the figures are modified, following recommendations of Referee #1 and #2.

*Another issue to point out is the fact that many are times that average period to study and plot is not indicated.*

In the process of revision we modified the plots. The information about averaging period is provided in the captions. We did not indicate it on curtain plots, not to overload the figures.

*Some specific details can be found below:*

*Page 1 line 16: How long are the 40 smoke episodes? Are they all equally representative? How many hours of measurement does each of them include? The lidar measures 24/7?*

The lidar was operated in the night time only. Duration of the measurements varied from 5 to 8 hours, depending on season. Corresponding comments is added to the manuscript.

*Page 2 line 45-48: It would be advisable to explain in detail that hygroscopic growth decreases the fluorescence capacity but does not affect the fluorescence spectrum.*

Particle hygroscopic growth leads to an increase in the backscattering coefficient, which in turn reduces the fluorescence capacity. However, how it was demonstrated by Veselovskii et al. (2023) and in current manuscript, the ratios of fluorescence backscattering in different channels do not depend on RH. Thus, the spectrum is not effected by hygroscopic growth. We added this comment to manuscript.

*Page 10 line 23: The text points that despite standard deviations, all measurement sessions reveal that the fluorescence backscattering gradually decreases with wavelength but Figure 8a fails to show that.*

Fluorescence backscattering gradually decreases with wavelength within the PBL. Corresponding comment is added to the manuscript.

*Page 14 Figure 12: The word "Backscattering" in the upper left subfigure should be corrected.*

Sorry, corrected.

*Page 16 line 322: Perhaps the average interval established to obtain the properties of aerosol particles is too long in time considering the extent and detail of the analysis of the events.*

Yes, but both urban and smoke layers are quite stable, so we think that this averaging well represents the mean values. The details of spatio-temporal variations of elastic and fluorescence backscattering are given by the curtain plots.

On balance, this paper represents a major advance in Raman and fluorescence-based lidar techniques, but the presentation of the work needs to be revised to maintain the high quality of the scientific content.